# Patients’ Satisfaction by SmileIn^TM^ Totems in Radiotherapy: A Two-Year Mono-Institutional Experience

**DOI:** 10.3390/healthcare9101268

**Published:** 2021-09-26

**Authors:** Giuditta Chiloiro, Angela Romano, Andrea D’Aviero, Loredana Dinapoli, Elisa Zane, Angela Tenore, Luca Boldrini, Mario Balducci, Maria Antonietta Gambacorta, Gian Carlo Mattiucci, Pierluigi Malavasi, Alfredo Cesario, Vincenzo Valentini

**Affiliations:** 1Istituto di Radiologia, Università Cattolica del Sacro Cuore, 00168 Rome, Italy; giuditta.chiloiro@policlinicogemelli.it (G.C.); luca.boldrini@policlinicogemelli.it (L.B.); mariaantonietta.gambacorta@policlinicogemelli.it (M.A.G.); vincenzo.valentini@policlinicogemelli.it (V.V.); 2UOC Radioterapia Oncologica, Dipartimento di Diagnostica per Immagini, Radioterapia Oncologica ed Ematologia, Fondazione Policlinico Universitario “A. Gemelli” IRCCS, 00168 Rome, Italy; angela.romano1@guest.policlinicogemelli.it (A.R.); loredana.dinapoli@gmail.com (L.D.); angelatenore2@gmail.com (A.T.); mario.balducci@policlinicogemelli.it (M.B.); giancarlo.mattiucci@policlinicogemelli.it (G.C.M.); 3UOS Psicologia Clinica, Fondazione Policlinico Universitario “A. Gemelli” IRCCS, 00168 Rome, Italy; 4Alta Scuola per l’Ambiente—ASA—Università Cattolica del Sacro Cuore, 25121 Brescia, Italy; elisa.f.zane@gmail.com (E.Z.); pierluigi.malavasi@unicatt.it (p.m.); 5Facoltà di Scienze della Formazione, Università Cattolica del Sacro Cuore, 25121 Brescia, Italy; 6Open Innovation, Fondazione Policlinico Universitario “A. Gemelli” IRCCS, 00168 Rome, Italy; alfredo.cesario@unicatt.it

**Keywords:** personalized medicine, best clinical practice, quality of care, radiotherapy, PREMs

## Abstract

Background: Patient’s satisfaction is recognized as an indicator to monitor quality in healthcare services. Patient-reported experience measures (PREMs) may contribute to create a benchmark of hospital performance by assessing quality and safety in cancer care. Methods: The areas of interest assessed were: patient-centric welcome perception (PCWP), punctuality, professionalism and comfort using the Lean Six Sigma (LSS) methodology. The RAMSI (Radioterapia Amica Mia SmileIN^TM (SI)^ My Friend Radiotherapy^SI^), project provided for the placement of SI totems with four push buttons using HappyOrNot technology in a high-volume radiation oncology (RO) department. The SI technology was implemented in the RO department of the Fondazione Policlinico Universitario A. Gemelli IRCCS. SI totems were installed in different areas of the department. The SI Experience Index was collected, analyzed and compared. Weekly and monthly reports were created showing hourly, daily and overall trends. Results: From October 2017 to November 2019, a total of 42,755 votes were recorded: 8687, 10,431, 18,628 and 5009 feedback items were obtained for PCWP, professionalism, punctuality, and comfort, respectively. All areas obtained a SI-approved rate ≥ 8.0. Conclusions: The implementation of the RAMSI system proved to be doable according to the large amount of feedback items collected in a high-volume clinical department. The application of the LSS methodology led to specific corrective actions such as modification of the call-in-clinic system during operations planning. In order to provide healthcare optimization, a multicentric and multispecialty network should be defined in order to set up a benchmark.

## 1. Introduction

Over recent years, the awareness of a patient’s perception of quality in healthcare systems has been increasing. Patient satisfaction is recognized as a key performance indicator to monitor the quality, not only of a single service, but also of the entire hospital [1]. Furthermore, patient satisfaction may influence perception of quality of care and, therefore, have a significant impact on treatment response [2,3,4,5]. Through systematic analysis of patient-relevant data, decision-making processes can be tailored to patients, empowering them to engage with healthcare systems, maximizing both their health and well-being.

Two main tools are advocated to capture patients’ perspectives on the effects of cancer care: Patient-Reported Outcome Measures (PROMs) and Patient-Reported Experience Measures (PREMs) [6]. While PROMs measure the impact of an illness and the effects of interventions in term of healthcare outcomes [7,8,9], PREMs assess a patient’s needs and experiences whilst receiving care [10,11].

PREMs are currently widely used and integrated into quality of care evaluation for chronic disease, as well as in emergency care services and for measuring a patient’s experience of medication use [12,13,14,15,16].

The focus on quality of life and care is becoming increasingly important in oncology as survival rates increase.

Cancer patients struggle not only with the disease but also with psychological burden, trauma, financial constraints, and many other forms of distress.

Among oncological treatments, radiotherapy is extremely stressful [17], because of its everyday burden on patients, which is often worsened by concomitant systemic therapies.

Patients can experience a loss of independence related to the duration of treatments (up to several weeks) and possible side effects [18]. Furthermore, patients may be hospitalized for a long period, interfering in their activities of daily living, especially when invasive surgical procedures are performed [17].

At the onset of RT, many patients experience reduced quality of life, pain and emotional distress [19]. PREMs can directly gather information from patients during treatments and capture their healthcare perspectives.

A comprehensive and aware consideration of the deficiencies identified by PREMs can lead to quality improvement (QI) interventions with significant consequences for patient quality of life. A better perception of the care approach also induces better adherence and compliance with a potential ameliorated impact on clinical outcomes as well.

The RAMSI (Radioterapia Amica Mia—SmileIN^TM (SI)^―My Friend Radiotherapy ^SI^) project foresaw the placement of SI totems with four push buttons using the HappyOrNot technology (RetailIN, Cesano Maderno (MB), Italy-https://smilein.it (access on 5 July 2021) in our RT department.

The RAMSI project enables the collection and analysis of patients feedback in the form of self-reported experience in real time. 

Healthcare services are based on complex and dynamic mechanisms that focus on patients and on improving the quality of care and services provided. Improving the quality of healthcare services has important implications for both the satisfaction of patients receiving care and the costs of services. In healthcare organizations, a widely used approach to maximize service quality and patient satisfaction is based on the Lean Six Sigma (LSS) methodology [20].

The LSS methodology was originally applied in several industries and subsequently applied, tested, and validated in the healthcare field [21]. The peculiarity of this field is that the patient is at the center of the mechanism and all attempts must be made to improve the care experience and thus to optimize processes. The LSS approach is structured according to the Define, Measure, Analyze, Improve and Control (DMAIC) method, which allows researchers to analyse and measure the problem and then carry out interventions that are monitored over time [20,22].

The main objective of this mono-institutional experience was to evaluate the usability of this technology in terms of simplicity and reproducibility to detect a patient’s empowerment and satisfaction during RT treatment. In this manuscript, the method and final metric of this experience are reported in terms of patients’ engagement in the four different investigated areas, according to the number of feedback items provided. 

The additional objective of our experience was to define a mono-institutional Radiation Oncology benchmark with which to compare the subsequent evaluations and implement healthcare services in a better way, applying the LSS approach through the DMAIC methodology.

## 2. Materials and Methods

### 2.1. Technology, Areas of Interest and Questions Choice

The technology was implemented in our radiotherapy department from October 2017 to November 2019. Physical SI totems were installed in the places of greatest affluence to promptly detect patient’s feedback and collect data on their experience during radiotherapy using HappyOrNot technology. Specifically, these locations were identified as: waiting rooms for clinics and treatment rooms; the access points and exit from the treatment rooms and the radiotherapy service.

Patients read the allocated question in the question sheet holder and gave their feedback anonymously by touching a smiley button. Four different faces define four assessment points: “very positive”, “positive”, “negative” and “very negative”.

In order to assess patients’ needs and experiences, four areas of interest related to PREMs were defined:patient-centric welcome perception (PCWP): the perception of human and environmental welcome during clinics and treatments;punctuality: visits and treatments time adherence to planned schedules;professionalism: healthcare workers competence or skill expected;comfort: environmental and human capability to accomplish patients’ needs.

In order to encourage the actual use of the system and thus obtain the highest number of feedback items, simple and understandable questions were chosen for the different areas and topics. The item creation team consisted of two radiation oncologists, two psychologists, one pedagogue and two technicians responsible for the company. In order to apply the DMAIC approach, the team defined the objective (D—define) which was firstly to validate the feasibility of the approach and then to implement corrective actions (I—improve) if necessary, monitored over time (C—control).

Through the reports for clinicians and patients, it was possible to obtain data (M—measure) which was then analyzed (A—analyze).

### 2.2. Reports for Clinicians

Patients’ feedback was recorded and reported in weekly and monthly data reports called “RAMSI Index”, which were sent via email to the dedicated team, the head physician, head nurse and chief of radiotherapy technician (RTT). A RAMSI data report with “Smile INdex” (SI Index) and “Smile In Approved” (SI Approved) was performed for each evaluated setting. Furthermore, an internal benchmark was obtained using the feedback received for the four areas after this preliminary observation period.

The SI Index was defined as the approval index with a value from 0 to 10, calculated as the weighted average of all the values obtained for each evaluation area.

The SI Approved, on the other hand, is a satisfaction rate ranging from 0% to 100% calculated monthly on the percentage of green smiles in relation to the total number of votes. An overall Approved SI was calculated on a monthly basis with a different graphical background to benchmark value: green or white if the value was superior or inferior to the benchmark value, respectively (Figure 1).

The benchmark was defined as SI Index reference related to the aggregated and homogeneous set of responses for each topic. As this experience represented, to our knowledge, a pilot study with no previous reference experience, the main focus was to collected data only to assess their value and time trend.

Following the feedback intercepted by this monitoring system and according to patients’ feedback, changes were made to both the formulation of questions and the positioning of the SI totems to collect increasingly precise data in relation to areas of possible service improvement.

The RAMSI system provides for the presence of a warning system that is activated if all the following parameters are exceeded during the considered time slot of one hour:at least five total responses in the time slot;at least three negative feedbacks in the time slot if the percentage of very negativity exceeds 30% of the the historical average percentage of very negative feedback of the previous 30 days (normalized average, days with 0 feedback are not considered);if the percentage of very negative feedback exceeds 25% of the total number of hourly responses.

### 2.3. Reports for Patients

The results shown to the patients by the RAMSI system vote are represented in two different ways. Inside the department, there is a notice board that depicts the monthly report of the different areas investigated by the SmileIN^TM^ totems. The data reported refer to PCWP, punctuality of the visits and therapies, received treatment quality, and environmental comfort. RAMSI’s trends can also be followed on the Facebook page.

Patients can also find the investigation tool on the “Fondazione Policlinico Universitario A. Gemelli IRCCS” website, also displayed on the notice board, through which it is possible to indicate if the search for information was successful. In order to deepen the patient’s perception, a section of the website has been reserved for suggestions and clarifications that can be sent anonymously and not influenced by the emotionality of the moment.

## 3. Results

The RAMSI project involved the placement of four SI totems within the radiotherapy division of the Fondazione Policlinico Universitario A. Gemelli IRCCS. During the observation period, from October 2017 to November 2019, a total number of 24,163 cumulative contacts were collected in RT.

Overall, more than 40,000 patients’ feedback items were obtained; 8687, 10,431, 18,628 and 5009 feedback items were obtained for PCWP, professionalism, punctuality, and comfort, respectively (Table 1). Validation of the items was an intrinsic objective of the study and was reached by patients’ feedback and the number of responses. No negative feedbacks were collected on the RAMSI technology and questions’ understanding.

The firstly proposed four questions were: “Did you feel welcomed as a person today?” for PCWP; “Did your appointment take place at the scheduled time?” for punctuality; “Have they been professional with you today?” for professionalism and “Are the environments comfortable?” for comfort.

Since May 2018, eight SI totems were installed in the radiotherapy division: three SI totems were designated for punctuality splitting three different questions: “Was the visiting hour respected?”, “Was your in-therapy visiting hour respected?” and “Did your treatment take place at the scheduled time?”; two SI totems for PCWP, other two for professionalism and two for comfort.

Systematic and periodical monitoring was performed in order to assess the trend of the different questions. The number of feedback items for each specific question, in absolute value such as the trends, were used to evaluate questions’ suitability.

### 3.1. Results of Questions Choice


*PCWP: “Did you feel welcomed as a person today?”*


Since the first installation, a SI totem has been placed in the main waiting room of the radiotherapy service. The trend throughout the installation period has remained consistent with a total of 6767 responses and an 80% positive response rating at present (5413 good and very good).

The PCWP survey was improved in May 2018 by the installation of an additional SI totem in the waiting room reserved for hospitalized and pediatric patients.

In both cases, at the analysis of the hourly pattern of responses, the bands of greatest significance were the service opening and closing times with a further peak at lunchtime.


*Punctuality: “Did your appointment take place at the scheduled time?”, “Was your clinical visit schedule respected?”, “Was the treatment schedule respected?” and “Was your in-therapy clinical visit schedule respected?”*


The first arrangement regarding the punctuality perception involved the placement of a SI totems within the service. Based on the evaluation of the trend of responses concerning the punctuality of the services, in September 2018 it was decided to divide the original question into two more specific ones: “Was your clinical visit schedule respected today?” and “Was the treatment schedule respected?” related to the service of outpatient visits and treatments, respectively.

The choice of having two different questions was to obtain a more precise definition of punctuality regarding visits and treatments separately, in order to perform specific efforts to improve the planning of healthcare services.

The decision to split the perception of the punctuality led to an overall increase in the perceived quality of punctuality and a better stratification of outpatient visits, clinical cases diversity, and treatments delivery.

A new SI totem describing perceived punctuality for “in-therapy visiting” has been implemented since May 2019 with a 75% approval rate.

Currently, the three SI totems are located in the treatment area and next to the main exit of the department. Overall, 18,628 feedback items about punctuality were recorded.


*Professionalism: “Have they been professional with you today?”*


A SI totem was placed in the treatment area at the first installation to detect the patient’s perception of the professionalism of health staff.

Since May 2018, a second SI totem has been positioned in the treatment area; the overall percentage of positive responses registered was 92%.


*Comfort: “Are the environments comfortable?”*


Two SI totems for assessing patient comfort have been placed in the premises of the treatment rooms since May 2018. The overall trend since installation is 90% (4508 positive responses recorded out of 5009 total feedback items).

### 3.2. Reports for Clinicians

The system generates weekly and monthly reports with time slots, service satisfaction rates, trends, according to time slots, days and with regard to each question, providing both fragmented and global data (Figure 1, Figure 2 and Figure 3).

An SI and SI Approved Indexes were available on the report with the correspondent benchmark value for each topic. A global index is also generated in the monthly report.

A mailing list system indicates peaks of negativity in the data for a specific time slot, allowing trends to be monitored in real time and the source of user dissatisfaction to be identified more precisely.

### 3.3. Reports for Patients

A monthly RAMSI Index is produced in order to identify the perception of patient satisfaction by combining data from the four investigated areas. These data are shared through a social media post containing a graphic representation that describes the results divided into the four survey areas (Figure 1).

The communication was also implemented through a clickable banner on the home page of the Fondazione Policlinico Universitario A. Gemelli IRCCS website that immediately shows the trend of the evaluation over the last month.

A main graph allows us to evaluate the monthly trend of the votes creating the median trend “SI Index”. For each question and area of interest, it is also possible to evaluate the daily progress at the specific time of the patients’ vote.

### 3.4. Warnings Report

Since the totems were installed, 48 warning events have occurred. Interestingly, it was observed retrospectively that the warning event, therefore, correlated to negative feedback, was always related to a real problem that had occurred. The warning issues in 38 cases (79.2%) were related to punctuality (visits and therapies), in five cases (10.4%) to environmental comfort and in another five cases (10.4%) to welcome perception.

### 3.5. Strategies for Improving the Service

The results of the RAMSI system surveys contributed to the implementation of specific corrective actions. Firstly, patients’ perception of punctuality contributed to the modification of the call-in-clinic system during clinical visits. We moved from a system that organized access based on the order in which patients arrived, to a system that takes into account the scheduled time for treatment. This corrective action has led to an improvement in satisfaction with punctuality.

In terms of comfort, a further corrective action was implemented: in autumn and winter, the temperature of areas of the department with higher access of patients was increased during off-peak hours.

Regarding professionalism and PCWP, the good performance encouraged us to continue the medical/technical and psychological training for nurses and RTTs with individual and group lessons/interventions.

## 4. Discussion

The current scenario is characterized by significant changes in the health service with a pressing need for new quality management of healthcare, both as a service and in terms of business management [23].

Radiotherapy is usually delivered on an outpatient basis, through cancer treatment centers, on a Monday–Friday schedule, starting from a single session up to an overall treatment time of 8 weeks. This intensive treatment period may provide a valuable screening and intervention opportunity to collect patients’ feedback over time and during the follow-up period. Patient satisfaction is currently used as an important indicator of healthcare quality and has been a widely explored concept, but still requires further research [24]. The available data are based on the consumer–producer relationship which seems reductive, being the satisfaction a multidimensional concept [25]. Satisfaction during the care pathway is critical, especially for the oncological patient undergoing long and often repeated treatments. In recent years, the patient’s perspective has become one of the fundamental pillars in determining hospital quality and the services offered.

Previous research has shown that several factors may affect patients’ perceptions of the quality of care.

These factors could be classified into the following two broad areas: person-related conditions and objective external conditions of care. Person-related conditions, such as patient gender [8,9], age [7,8], educational level [8,9], physical health and psychological well-being [10], and whether patients are hospitalized as emergencies or with planned admissions [8,9], are related to the patients’ perception of the quality of care [26].

Historically, patient satisfaction has always been measured through tools such as surveys and questionnaires.

The RAMSI project effectively puts the patient at the center of the therapeutic process as a person in her/his complexity in order to preserve their QoL and human dignity during the radiation treatment. Furthermore, it provides a fast, quick and easy-to-use tool to extract patient satisfaction data. The SI totems are characterized by easy interaction and understanding, as they are based on four buttons with colored smiles. This intuitive interaction makes this technology usable by patients of different ages, education levels, and gender, ensuring a large collection of feedbacks in a simple, free, quick, anonymous and spontaneous way. The RAMSI project also aimed to improve the services offered in a division of RT to improve the experience of care. The idea was to use the “happy or not” method in the healthcare setting to first assess its feasibility and then measure the healthcare experience. By intercepting any limitations or mismanagement of the care process, the type of technology used can promote rapid identification and resolution of problems in real time and promote long-term corrective actions.

The RAMSI study embraces this complexity through a multi-professional approach that involves several professionals to address quality perception, analyzing four different areas in order to investigate in depth how to solve potential problems.

This approach is successful due to the high number of votes collected every month; furthermore, the large number of feedback items collected (over 40,000) in a high-volume clinical department, underlines the usability of the system to perform changes in daily practice according to patients’ needs in order to ensure a high-quality health service. One of the strengths of the project was the possibility to have anonymous aggregated data that are easy to find and collect, from which information can be obtained. The automation of the SI totems produces automated reporting that is easy to communicate and share and it is highly comprehensible to different stakeholders.

These reports may increase the perception of the service provided and received for both clinicians and patients.

This allows a quick and easy evaluation of the service provided. This was carried out, during the first year of the project, by analyzing the collected data and then comparing them with the actual data. This methodology led to corrective action, trying to remove the causes of possible negative feedback. It was possible due to the large amount of data available and because the negative data always directly correlated with a “warning” event that actually occurred. The improvement of services was achieved through a reorganization of the time slots dedicated to medical visits, changes in the management of patient arrivals and the waiting room, and the opening of more outpatient clinics. Measuring the values over time, before and after the corrective measures, will help to assess whether they were valid and effective. The application of the LSS methodology proved feasible and effective in detecting problems related to the patient experience in a radiotherapy department. Although weaknesses in the LSS methodology have been observed, the benefits of applying this methodology have been demonstrated in the literature. The implementation of the LSS approach has proven to be effective in reducing healthcare service costs and errors, through the steps of the DMAIC methodology, which allows us, step by step, to identify a problem and improve performance. The areas in which the methodology has been applied range from the optimization of treatment times in emergency rooms and discharge times in emergency departments, to waiting times, delivery of medical reports, unnecessary medical costs and the reduction of errors of medical reports, unnecessary medical costs, and so on [20,22,27,28]. This significant participation is also a useful tool for clinicians to guide their actions; and, as a result of our experience, RAMSI guided implementations have led to higher patients’ accomplishment, as demonstrated by the “punctuality score” increase by taking into account appointment booking time and sequentially changing the call-in-clinic system.

The soundness of these results supports the hypothesis that a deeper consideration of the relational needs of the patient could lead to better care practice and stronger cooperation between patients and healthcare providers, especially in the cancer care setting.

Furthermore, the evaluation of the efficiency and effectiveness of the procedures relating to the defined operational objectives offers the possibility to translate the non-compliance with certain processes into corrective and preventive optimization actions.

From a psychological point of view, a limitation could be related to the occasional emotional experience being limited to certain inefficiencies or isolated conflict situations. These negative feedback items could have an important impact on one or more variables in terms of negative peaks. However, the detection of patient dissatisfaction can serve as a useful element to improve services and care in a focused manner.

Logistically, a limitation is linked to the possibility for anyone to vote, even if not interested in the topic.

## 5. Conclusions

The RAMSI project represents, to the best of our knowledge, the first reported application of SI in clinical practice, and specifically in cancer care, to investigate patient’s perception of the quality of services and care received.

In relation to the uniqueness of the experience, there was no validated benchmark for the results of the analysis.

Our experience should itself represent a first national benchmark for the Italian radiation oncology community and the starting point for the creation of a multicentric and multi-specialty benchmark.

The evaluation of health facilities through the RAMSI may allow a definition of the quality of assistance and the exploration of gray zones in patients’ care in daily practice in order to optimize clinical procedures and treatments.

## Figures and Tables

**Figure 1 healthcare-09-01268-f001:**
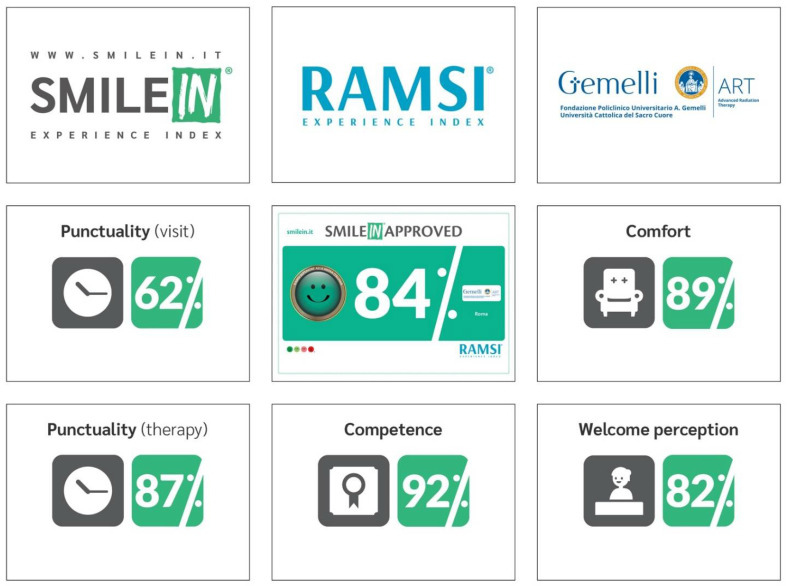
Monthly SmileIN approved.

**Figure 2 healthcare-09-01268-f002:**
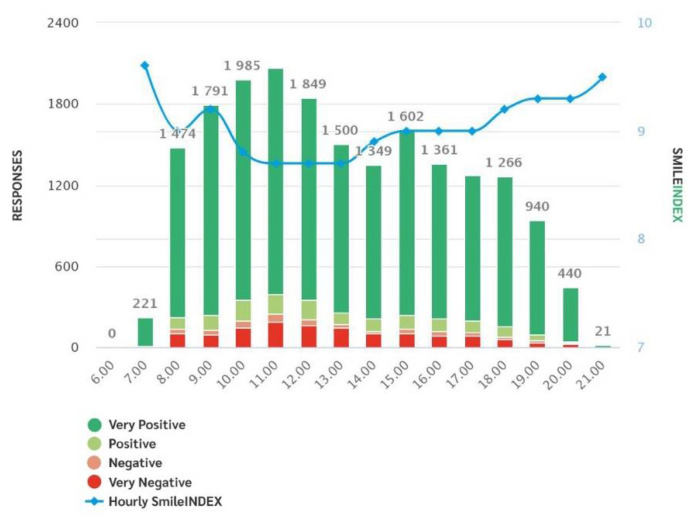
Trends by hour.

**Figure 3 healthcare-09-01268-f003:**
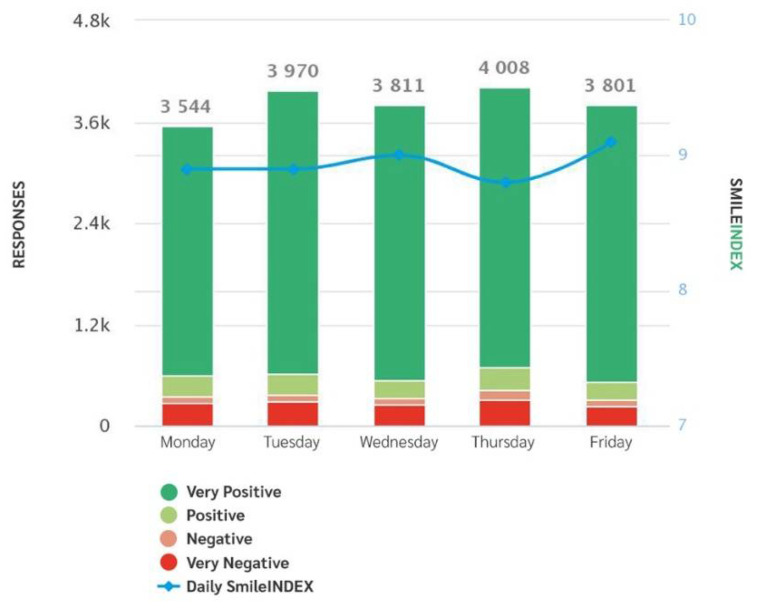
Trends by days.

**Table 1 healthcare-09-01268-t001:** Overall survey evaluation.

PREMs	Location	Collection Period	SMILE-Index	SMILE-IN Approved	Best Timeframe	Worst Timeframe	Best Performance	Worst Performance	Total Feedbacks
PATIENT CENTRIC WELCOME PERCEPTION		“Did you feel welcomed as a person today?”							
	Main waiting room	10/2017–ONGOING	7.8	80%	8 a.m.	6 p.m.	-	-	6767
	Internal waiting room	05/2018–ONGOING	8.6	87%	8 p.m.	9 p.m.	-	-	1920
	Total	10/2017–ONGOING	8.0	82%	8 a.m.	6 p.m.	8.7	7.8	8687
PROFESSIONALISM		“Have they been competent with you today?”							
	Main waiting room	10/2017–1/2018	9.0	91%	3 p.m.	9 p.m.	-	-	4237
	Entrance bunker	05/2018–ONGOING	9.3	93%	9 a.m.	9 p.m.	-	-	3442
	Treatment area main room	11/2018–ONGOING	9.1	92%	1 p.m.	10 a.m.	-	-	2772
	Total	10/2017–ONGOING	9.1	92%	3 p.m.	9 p.m.	9.3	9.0	10,431
PUNCTUALITY		“Was your appointment schedule respected today?”							
	Main waiting room	10/2017–06/2018	5.0	51%	7 a.m.	1 p.m.	-	-	4454
	Waiting room 2	05/2018–06/2018	6.8	70%	7 a.m.	9 p.m.	-	-	132
	Exit 1	06/2018–09/2018	6.0	60%	7 a.m.	9 p.m.	-	-	2005
	Exit 2	06/2018–09/2018	5.5	55%	8 a.m.	9 p.m.	-	-	887
	Total	10/2017–09/2018	5.4	54%	8 a.m.	9 p.m.	6.8	5.0	7478
PUNCTUALITY		“Was your in-therapy clinical visit schedule respected?”							
	Waiting room 2	05/2019–ONGOING	7.4	75%	7 a.m.	8 p.m.	-	-	1084
PUNCTUALITY		“Was the treatment schedule respected?”							
	Treatment area exit	09/2018–11/2018	7.3	74%	8 a.m.	7 a.m.	-	-	2046
	Treatment area main room	11/2018–ONGOING	8.4	85%	7 a.m.	9 p.m.	-	-	3069
	Total	09/2018–ONGOING	8.0	81%	8 a.m.	9 p.m.	8.4	7.3	5115
PUNCTUALITY		“Was your clinic visit schedule respected?”							
	Clinical visit exit	09/2018–ONGOING	6.0	61%	8 a.m.	7 a.m.	6.3	6.0	4951
COMFORT		“Are the environments comfortable?”							
	Treatment area main room	05/2018–11/2018	8.9	90%	5 p.m.	9 p.m.	-	-	955
	Waiting room 2	05/2018–ONGOING	8.9	90%	9 a.m.	4 p.m.	-	-	4054
	Total	05/2018–ONGOING	8.9	90%	9 a.m.	4 p.m.	8.9	8.9	5009

## Data Availability

The data presented in this study are available on request from the corresponding author.

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
