# Peer review of "Patients’ Satisfaction by SmileInTM Totems in Radiotherapy: A Two-Year Mono-Institutional Experience"

_healthcare, 2021, doi:10.3390/healthcare9101268_

Round 1

Reviewer 1 Report

Thank you very much for giving me this opportunity to review this article. I found some major issues that should be addressed before publication. 

  1. The abstract is not according to the journal formate.
  2. The significance and motivation is unclear to me in introduction section. Why and how this study is being investigated. what are the theoretical and practical implications. The research gap is unclear to the reader. 
  3. I suggest to write in small paragraphs not like two lines.
  4. I did not found anything related to the literature. Without literature review - every study is incomplete. Please add this part.
  5. Where are the theorectical and practical implications.?
  6. Please add limitation and future research section after the conclusion. 
  7. There are so many grammatical errors. I strongly suggest to do proofreading from english native.  

Good Luck

Author Response

1. The abstract is not according to the journal formate.

Thanks, we modified abstract.

2. The significance and motivation is unclear to me in introduction section. Why and how this study is being investigated. what are the theoretical and practical implications. The research gap is unclear to the reader.

Thank you for pointing this out. We decided to rephrase the main objective of the study, considering the HappyOrNot technology as a tool to collect information from patients that can then influence healthcare decisions in order to improve the services offered to patients.

3. I suggest to write in small paragraphs not like two lines.

Thank you for your suggestion, we have made the requested modification.

4. I did not found anything related to the literature. Without literature review - every study is incomplete. Please add this part.

Thanks for pointing this out. This is, to our knowledge, the first documented experience of implementing HappyOrNot technology in hospitals as a tool to collect data and use them for healthcare interventions in radiotherapy. We have made this explicit in the discussion.

5. Where are the theorectical and practical implications.?

This type of technology makes it possible to have a large amount of anonymous data from patients that can be analysed over time. Any changes in patient feedback can be correlated with recorded events. This information can be a starting point for implementing corrective measures and subsequently validating them within the RT department. This type of approach could potentially be applied in other departments in order to improve healthcare.

6. Please add limitation and future research section after the conclusion. 

We have described the limitations of the study in the conclusion section, as suggested by the reviewer. As the RAMSI project is a preliminary experience on the use of this new technology, we believe that validation is also needed in other RT departments of other national and international hospitals. Therefore, the same model could be applied in other healthcare settings in the future.

7. There are so many grammatical errors. I strongly suggest to do proofreading from english native.  

We provided revision by native speaking.

Reviewer 2 Report

This article is reporting on two years of experience with a new system to assess patients experience with a radiotherapy service using local totems with a simple scoring system using smileys at four levels.

It is an interesting report on a new experience, but it is difficult to find any research or evidence for the system or any information about the choices before and during the observation period.

The main idea according to the authors is that “measures (PREMs) may contribute to create a benchmark of hospital performance by assessing quality and safety in cancer care.” This will be done by registering “patient centric welcome perception (PCWP), punctuality, professionalism and comfort “. One of the major problems is that this coupling is not evident and not made in the article and the construct and content validity of the measures of the four areas are not reported here.

Thus the authors overall need to be very clear about what this article reports on and what it does not.

For the reviewer it is a case report or at largest a feasibility study in relation to how the measures are used. Nevertheless, it needs to be clear how this is assessed as well. No interviews, comparisons to other measures, focus groups?  

Maybe it is better to present it as a way to inform health professionals and their organizations with respect to workflow optimization. To the reviewer this is the real and most important focus and learning point from the article.

Specific comments:

Introduction – the authors describe the distress in relation to cancer and radiotherapy, but this relates to PROMS and not PREMs. The introduction should focus more on the PREM perspective and directly argue for the four areas addressed in the PREM measure.

The authors state the following: “The main objective of this mono-institutional experience was to evaluate the usability of this technology in terms of simplicity and reproducibility to detect patient’s empowerment and satisfaction during RT treatment. In this manuscript, the method and the final metric of this experience are reported in terms of patients’ engagement in the four different investigated areas, according to the number of provided feedbacks.” This should be re-scoped and it should be clearer with respect to what the authors want to report on and what their metrics actually allow them to conclude. The usage by itself does not inform about how many patients, or which group of patients, actually report or whether it is the patients or their relatives following them. Why is the number of reports a useful measure? Would it not be better to just report on how the data may inform and influence the work of the health professionals and their organization? This would also better align with the conclusion.

Material and methods:

It is unclear how the questions belonging to each domain was developed and tested. How was the interpretation checked? Cognitive testing? How was the items validated with respect to overlap or validity? Also, is it correct that the scale had the most positive response first and the less favorable at last? This is opposite many other scoring systems. Please provide more information in relation to the development of the items and the cognitive and validity testing.

Results

It would be of interest to know how many contacts and how many patients have passed through the department in the observation period to understand the actual percentage of answers.

Would be great to have more information about the decisions and reflections when the questions were changed.

Discussion:

Good to see that the focus is on satisfaction and influence on the way work is organized. The reviewer is concerned about the following sentence “… and understanding, as they are based on four buttons with colored smiles. This intuitive interaction makes this technology usable by patients of different ages, education levels, and gender, ensuring a large collection of feedbacks in a simple, free, quick, anonymously”.  Intuitively it seems correct, but what is the evidence, should be omitted as the authors do not know whether people who are elder or lower educated tends to use this less or higher, or educated do not want to use such a simple way for feedback.

Another concern is the use of the term “usability” The authors write “the large number of feedbacks collected (over 40.000 patients’ feedbacks) in a high-volume clinical department, underlines the usability of the system”. This high number of inputs does not tell anything about usability. Maybe feasibility if the authors can document that the users are evenly distributed with respect to age, sex, education and cultural background.

And if the journal has a section with case reports, this article may suit there, but it does not fit into a research article.

Author Response

This article is reporting on two years of experience with a new system to assess patients experience with a radiotherapy service using local totems with a simple scoring system using smileys at four levels.

It is an interesting report on a new experience, but it is difficult to find any research or evidence for the system or any information about the choices before and during the observation period.

The main idea according to the authors is that “measures (PREMs) may contribute to create a benchmark of hospital performance by assessing quality and safety in cancer care.” This will be done by registering “patient centric welcome perception (PCWP), punctuality, professionalism and comfort “. One of the major problems is that this coupling is not evident and not made in the article and the construct and content validity of the measures of the four areas are not reported here.

Thus the authors overall need to be very clear about what this article reports on and what it does not.

For the reviewer it is a case report or at largest a feasibility study in relation to how the measures are used. Nevertheless, it needs to be clear how this is assessed as well. No interviews, comparisons to other measures, focus groups?  

Maybe it is better to present it as a way to inform health professionals and their organizations with respect to workflow optimization. To the reviewer this is the real and most important focus and learning point from the article.

Specific comments:

Introduction – the authors describe the distress in relation to cancer and radiotherapy, but this relates to PROMS and not PREMs. The introduction should focus more on the PREM perspective and directly argue for the four areas addressed in the PREM measure.

We have now focused the introduction on PREMs.

The authors state the following: “The main objective of this mono-institutional experience was to evaluate the usability of this technology in terms of simplicity and reproducibility to detect patient’s empowerment and satisfaction during RT treatment. In this manuscript, the method and the final metric of this experience are reported in terms of patients’ engagement in the four different investigated areas, according to the number of provided feedbacks.” This should be re-scoped and it should be clearer with respect to what the authors want to report on and what their metrics actually allow them to conclude. The usage by itself does not inform about how many patients, or which group of patients, actually report or whether it is the patients or their relatives following them. Why is the number of reports a useful measure? Would it not be better to just report on how the data may inform and influence the work of the health professionals and their organization? This would also better align with the conclusion.

We would like to thank the reviewer for the comment. We totally rephrased the main objective of the study, considering crucial the HappyOrNot technology as a tool to collect information obtained from patients. These information would then be used to achieve the additional objective of the study, which is the application of the LSS approach for the implementation of healthcare services in the hospital.

Material and methods:

It is unclear how the questions belonging to each domain was developed and tested. How was the interpretation checked? Cognitive testing? How were the items validated with respect to overlap or validity? Also, is it correct that the scale had the most positive response first and the less favorable at last? This is opposite many other scoring systems. Please provide more information in relation to the development of the items and the cognitive and validity testing.

The study methodology did not included a validation for the items.

Results

It would be of interest to know how many contacts and how many patients have passed through the department in the observation period to understand the actual percentage of answers.

Thanks for the comment. Unfortunately, the data concerning the number of patients is not available because the management system of the RT department is designed to record only contacts (and not the number of patients) in a given period of time. The available data is only related to the number of patients per day, which is on average about 250.

Would be great to have more information about the decisions and reflections when the questions were changed.

 Thanks, we added further informations.

Discussion:

Good to see that the focus is on satisfaction and influence on the way work is organized. The reviewer is concerned about the following sentence “… and understanding, as they are based on four buttons with colored smiles. This intuitive interaction makes this technology usable by patients of different ages, education levels, and gender, ensuring a large collection of feedbacks in a simple, free, quick, anonymously”.  Intuitively it seems correct, but what is the evidence, should be omitted as the authors do not know whether people who are elder or lower educated tends to use this less or higher, or educated do not want to use such a simple way for feedback.

The term usability has been removed, as suggested by the reviewer. We have outlined this as a limitation of the study in the conclusions.

Another concern is the use of the term “usability” The authors write “the large number of feedbacks collected (over 40.000 patients’ feedbacks) in a high-volume clinical department, underlines the usability of the system”. This high number of inputs does not tell anything about usability. Maybe feasibility if the authors can document that the users are evenly distributed with respect to age, sex, education and cultural background.

We preferred to use the term applicability, as both usability and feasibility are inappropriate terms for the methodology of the study.

Reviewer 3 Report

Dear Authors,

the article deals with an interesting and very topical issue. However, there are some limitations regarding 
(a) the clarity in the exposition of concepts. In particular, the methodological section and that devoted to the results seem to be mixed and the reader finds it difficult to distinguish between them. It could be useful to add a  description of the procedures and of the different phases of the research within the Method section. The article could benefit from the introduction of a sub-section entitled "Procedures", but this is not compulsory;
(b) the links between results and implemented actions, that are not immediately understandable;
(c) the limited number of bibliographical references, some of which are quite outdated. For some sections of the article I have added some bibliographic suggestions;
(d) the excessive use of acronyms, which are not always made explicit;

(e) the results, which are only partially argued. No assumptions are made about the reasons for improvements or deteriorations over time with respect to the dimensions investigated. Figure 4 is extremely interesting but is not mentioned in the body of the text. Its data are not commented on. Only a part of table 1 is readable.

Suggestions

Abstract 

SI stands for Smile, I presume, but it might be better to explain.

Introduction

Lines 52-53. Perhaps it might be useful for the authors to refer to some other previous and recent work on the subject. For example: 

Nardin, S., Mora, E., Varughese, F. M., D'Avanzo, F., Vachanaram, A. R., Rossi, V., ... & Gennari, A. (2020). Breast cancer survivorship, quality of life, and late toxicities. Frontiers in Oncology, 10, 864.

Invernizzi, M., Kim, J., & Fusco, N. (2020). Quality of Life in Breast Cancer Patients and Survivors. Frontiers in Oncology, 10.

Lines 54-55: Bibliographic references could be added. With regard to Italian patients, the following could be mentioned:

Cormio, C., Caporale, F., Spatuzzi, R. et al. Psychosocial distress in oncology: using the distress thermometer for assessing risk classes. Support Care Cancer 27, 4115–4121 (2019). https://doi.org/10.1007/s00520-019-04694-4

Muzzatti, B., Bomben, F., Flaiban, C. et al. Quality of life and psychological distress during cancer: a prospective observational study involving young breast cancer female patients. BMC Cancer 20, 758 (2020). https://doi.org/10.1186/s12885-020-07272-8

Civilotti, C., Maran, D. A., Santagata, F., Varetto, A., & Stanizzo, M. R. (2020). The use of the Distress Thermometer and the Hospital Anxiety and Depression Scale for screening of anxiety and depression in Italian women newly diagnosed with breast cancer. Supportive Care in Cancer, 1-8.

Line 72: The acronym RT is used without prior explanation. Although it is clear to the experts in the field, the first time it is used it would be better to precede it with the use of the full term.

Line 64 and line 69: Round brackets for bibliographical references should be replaced in the text by square brackets (see bibliographical references 21 and 22). Some of the bibliographical references are dated. For example, in this case, although still limited to the Italian situation, the following one could be cited:

De Rosis, S., Cerasuolo, D., & Nuti, S. (2020). Using patient-reported measures to drive change in healthcare: the experience of the digital, continuous and systematic PREMs observatory in Italy. BMC health services research, 20(1), 1-17.

Line 80: There is probably something wrong with the number of bibliographical references.  [20] follows references (21) and (22). However, these numbers are probably incorrect. In fact, references [21] and [22] also follow reference [20] and are related to the content under discussion in that part of the text. 

It would be better to add some more explanation of what Lean Six Sigma is. Not all readers may be familiar with this method.

Furthermore, the authors could add more recent bibliographical references regarding the use of LSS in the healthcare systems and/or specific contexts. For example

Henrique, D. B., & Godinho Filho, M. (2020). A systematic literature review of empirical research in Lean and Six Sigma in healthcare. Total Quality Management & Business Excellence, 31(3-4), 429-449.

Arcidiacono, G., & Pieroni, A. (2018). The revolution lean six sigma 4.0. Int. J. Adv. Sci. Eng. Inf. Technol, 8(1), 141-149.

Line 89: the authors use the term technology. If they refer to PREMS and/or to SI totems and/or HappyOrNot technology, and not to LSS, it might be better to make this explicit, since in the previous lines they were presenting LSS.

Materials and Methods

Lines 142-145: general statements. A more detailed description would be appreciated (after how long and after how many patients' responses were the changes made?). This phase is something similar to a pre-test and these data are important.

Results

Some information might move from this section to the methodological one. Specifically, lines: 167-168; 173-187. These are not results but procedures. 

Line 180: SI appears as an apex.

Lines 194-195:  For the first time a reference to pediatric patients appears. It would be appreciated if the authors would write, at least in a footnote, who used the TOTEM (the child? the parent?). In addition, it would have been interesting to see whether there were significant differences between adults and children, as well as with regard to other socio-demographic or disease-related variables. The reader at this point might wonder why no socio-demographic data (e.g. gender, age, level of schooling, etc.) or data related to the type of disease were collected. Perhaps, in the methodological section, the reason for this choice can be made explicit. Furthermore, how is it ensured that the answers are only given by patients and not also by caregivers (for example)? 

Line 203 and following: 

Capital letter for “the”.

On lines 180-183, the authors talk about May 2018 and they state that they have divided the initial question on punctuality into three different questions. Here (lines 204-208) they refer instead to September 2018 and talk about 2 different questions. Furthermore (lines 215-216), the authors refer to a new SI totem and to a new question. It is all rather confusing. 

There are two problems: the first is related to the fact that these data should be discussed in a subsection (procedures) of the method section and not among the results, since they are not results; the second is related to the fact that the reader is confused by non-uniform information.

Line 220: Also at this point, the authors seem to mix procedures and results. Furthermore, while for “Comfort” the authors present percentages and occurrences (these last within the round brackets), for “Professionalism” they do not give this datum. It is all rather confusing.

Lines 227-229: The italics should be replaced with normal. 

Formal error: The words "trends" (line 232) and “according” (line 233) are on two different lines. 

Lines 241 and following: Once again the authors seem to mix procedures and results. What they write here is similar to what they wrote above (see lines 154-164).

Lines 264 and following: It is not clear how corrective actions were chosen, especially those related to comfort. Why focus on temperature? By reading what precedes in the text, it does not seem that this information was known. The reader has the right to ask why other improvement actions were not chosen. For example, why not light the rooms differently or paint them in a different color or buy more comfortable armchairs? If, on the contrary, the indication of temperature emerged from the patients' answers, this should be made explicit above. 

Line 269The acronym RTTs appears for the first time. Although it is clear that it refers to radiotherapists, it would be better to use the full term. 

Discussion

Line 274: Bibliographic references could be updated. The authors refer in this case [reference n. 23] to a work of twelve years ago in relation to the "current situation".

Line 281: The “consumer-producer” dichotomy (which appears here for the first time) does not seem to be relevant without an adequate bibliographic reference to justify its use. 

Line 285Since, according to the authors, the patients’ perspective is a “pillars" in determining hospital quality, why they do not refer to recent studies on the topic? 

Line 291 and followingSince in this study the mentioned variables are not taken into account, this should appear as one of the limitations.

Line 297: Although the acronym QoL is clear, it should be explained. Furthermore, since (like QI), it is used only once, it would be better to replace it with the full term, also in view of the numerous acronyms the authors use in the article.

Lines 334-336: general statements without adequate supporting bibliographic references.

Line 347: delete the dot at the beginning of the line. It is not clear why the authors refer to "relational needs" only at the end of the article. What is their relationship with the aspects assessed through the totems (punctuality, comfort, PCWP, and professionalism)?  

Lines 358-359Another limitation, contrary to the one indicated by the authors, could be that the same person votes twice.

All the best for your paper.

Author Response

Abstract 

SI stands for Smile, I presume, but it might be better to explain.

Thanks for the comments, we used SI abbreviation for capital letter of SmileIn.

Introduction

Lines 52-53. Perhaps it might be useful for the authors to refer to some other previous and recent work on the subject. For example: 

Nardin, S., Mora, E., Varughese, F. M., D'Avanzo, F., Vachanaram, A. R., Rossi, V., ... & Gennari, A. (2020). Breast cancer survivorship, quality of life, and late toxicities. Frontiers in Oncology, 10, 864.

Invernizzi, M., Kim, J., & Fusco, N. (2020). Quality of Life in Breast Cancer Patients and Survivors. Frontiers in Oncology, 10.

Thanks for the comment, we added citations.

Lines 54-55: Bibliographic references could be added. With regard to Italian patients, the following could be mentioned:

Cormio, C., Caporale, F., Spatuzzi, R. et al. Psychosocial distress in oncology: using the distress thermometer for assessing risk classes. Support Care Cancer 27, 4115–4121 (2019). https://doi.org/10.1007/s00520-019-04694-4

Muzzatti, B., Bomben, F., Flaiban, C. et al. Quality of life and psychological distress during cancer: a prospective observational study involving young breast cancer female patients. BMC Cancer 20, 758 (2020). https://doi.org/10.1186/s12885-020-07272-8

Civilotti, C., Maran, D. A., Santagata, F., Varetto, A., & Stanizzo, M. R. (2020). The use of the Distress Thermometer and the Hospital Anxiety and Depression Scale for screening of anxiety and depression in Italian women newly diagnosed with breast cancer. Supportive Care in Cancer, 1-8.

Thanks for the comment, we added citations.

Line 72: The acronym RT is used without prior explanation. Although it is clear to the experts in the field, the first time it is used it would be better to precede it with the use of the full term.

Thanks for the comment, we provided clarification.

Line 64 and line 69: Round brackets for bibliographical references should be replaced in the text by square brackets (see bibliographical references 21 and 22). Some of the bibliographical references are dated. For example, in this case, although still limited to the Italian situation, the following one could be cited:

De Rosis, S., Cerasuolo, D., & Nuti, S. (2020). Using patient-reported measures to drive change in healthcare: the experience of the digital, continuous and systematic PREMs observatory in Italy. BMC health services research, 20(1), 1-17.

Thanks for the comment, we corrected citations and added the reference suggested.

Line 80: There is probably something wrong with the number of bibliographical references.  [20] follows references (21) and (22). However, these numbers are probably incorrect. In fact, references [21] and [22] also follow reference [20] and are related to the content under discussion in that part of the text. 

Thanks for the comment, we corrected citations.

It would be better to add some more explanation of what Lean Six Sigma is. Not all readers may be familiar with this method.

Thank you for your comment. We have added a statement in the introduction to define what the LSS approach is and two bibliographic references as follow:

“The LSS methodology was created with the aim of improving the quality of the production process by eliminating errors and reducing costs. Lean thinking aims to reduce waste by minimizing the variation of processes to a capacity of ± six standard deviations (sigma)”

Furthermore, the authors could add more recent bibliographical references regarding the use of LSS in the healthcare systems and/or specific contexts. For example

Henrique, D. B., & Godinho Filho, M. (2020). A systematic literature review of empirical research in Lean and Six Sigma in healthcare. Total Quality Management & Business Excellence, 31(3-4), 429-449.

Arcidiacono, G., & Pieroni, A. (2018). The revolution lean six sigma 4.0. Int. J. Adv. Sci. Eng. Inf. Technol, 8(1), 141-149.

Thanks for the comment, we added citations.

Line 89: the authors use the term technology. If they refer to PREMS and/or to SI totems and/or HappyOrNot technology, and not to LSS, it might be better to make this explicit, since in the previous lines they were presenting LSS.

Thanks for the comment, we rephrased the paragraph.

Materials and Methods

Lines 142-145: general statements. A more detailed description would be appreciated (after how long and after how many patients' responses were the changes made?). This phase is something similar to a pre-test and these data are important.

Thank you for your comment. We clarified after how many months of the pre-test phase corrective actions were planned.

Results

Some information might move from this section to the methodological one. Specifically, lines: 167-168; 173-187. These are not results but procedures. 

Thank you for pointing this out. We have made the change as requested by the reviewer.

Line 180: SI appears as an apex.

Thanks for the comment, we corrected it.

Lines 194-195:  For the first time a reference to pediatric patients appears. It would be appreciated if the authors would write, at least in a footnote, who used the TOTEM (the child? the parent?). In addition, it would have been interesting to see whether there were significant differences between adults and children, as well as with regard to other socio-demographic or disease-related variables. The reader at this point might wonder why no socio-demographic data (e.g. gender, age, level of schooling, etc.) or data related to the type of disease were collected. Perhaps, in the methodological section, the reason for this choice can be made explicit. Furthermore, how is it ensured that the answers are only given by patients and not also by caregivers (for example)? 

Thanks for the clarification. It would certainly be interesting, but at present the limitation of technology (which we have also made explicit in the discussion) does not allow us to extrapolate this kind of information.

We report the change to the manuscript as follows:

“Furthermore, patient-related factors (e.g. age, level of education and physical well-being, etc.) were not included in the data we analysed, as the feedback were anonymous and the totems were available to all patients.”

Line 203 and following: 

Capital letter for “the”.

Thanks for the comment, we corrected it.

On lines 180-183, the authors talk about May 2018 and they state that they have divided the initial question on punctuality into three different questions. Here (lines 204-208) they refer instead to September 2018 and talk about 2 different questions. Furthermore (lines 215-216), the authors refer to a new SI totem and to a new question. It is all rather confusing. 

Thank you for your comment. We have clarified in the materials and methods section the introduction of totems over time and how the questions have been modified as follows:

“Since May 2018, 8 SI totems were installed in the Radiotherapy division: 3 SI totems were designated for punctuality splitting 2 different questions: “Was the visiting hour respected?” and “Did your treatment take place at the scheduled time?” and a new totem was subsequently installed in September 2019 with the question: "Was your in-therapy visiting hour respected?"; 2 SI totems for PCWP, other 2 for professionalism and 1 for comfort. Systematic and periodical monitoring was performed in order to assess the trend of the different questions. The number of feedbacks for each specific question, in absolute value such as the trends, were used to evaluate questions suitability.”

There are two problems: the first is related to the fact that these data should be discussed in a subsection (procedures) of the method section and not among the results, since they are not results; the second is related to the fact that the reader is confused by non-uniform information.

As suggested, we have moved these sentences to the methods section and better explained them according to the previous comment.

Line 220: Also at this point, the authors seem to mix procedures and results. Furthermore, while for “Comfort” the authors present percentages and occurrences (these last within the round brackets), for “Professionalism” they do not give this datum. It is all rather confusing.

As suggested by the reviewer, we have organised the materials and methods and results sections.

Lines 227-229: The italics should be replaced with normal. 

Thanks for the comment, we corrected it.

Formal error: The words "trends" (line 232) and “according” (line 233) are on two different lines. 

Thanks for the comment, we corrected it.

Lines 241 and following: Once again the authors seem to mix procedures and results. What they write here is similar to what they wrote above (see lines 154-164).

As suggested by the reviewer, we reorganised the manuscript as for the previous sections.

Lines 264 and following: It is not clear how corrective actions were chosen, especially those related to comfort. Why focus on temperature? By reading what precedes in the text, it does not seem that this information was known. The reader has the right to ask why other improvement actions were not chosen. For example, why not light the rooms differently or paint them in a different color or buy more comfortable armchairs? If, on the contrary, the indication of temperature emerged from the patients' answers, this should be made explicit above. 

Thank you for your comment. The corrective action on the room temperature was carried out deductively based on the seasonal temperature change (autumn/winter) and the negative feedback at less crowded time slots. We modified the section as follow:

“In terms of comfort, there was an increase in negative feedback in the early morning and evening hours when the RT areas were not very frequented. This allowed us to deduce a problem related to the room air-conditioning, so we implemented a corrective measure that improved the comfort feedback in the following weeks”

Line 269The acronym RTTs appears for the first time. Although it is clear that it refers to radiotherapists, it would be better to use the full term. 

Thanks for the comment, we corrected with RTT as previously mentioned.

Discussion

Line 274: Bibliographic references could be updated. The authors refer in this case [reference n. 23] to a work of twelve years ago in relation to the "current situation".

Following the reviewer's advice, we have added two more recent citations, including one review, describing new models of patient-centred health.

Line 281: The “consumer-producer” dichotomy (which appears here for the first time) does not seem to be relevant without an adequate bibliographic reference to justify its use. 

Thanks for the advice, we have included a reference highlighting the relationships in existing producer-driven and buyer-driven governance structures in the digital era.

Line 285Since, according to the authors, the patients’ perspective is a “pillars" in determining hospital quality, why they do not refer to recent studies on the topic? 

We have added a recent literature reference investigating the role of patient experience in hospitals.

Line 291 and followingSince in this study the mentioned variables are not taken into account, this should appear as one of the limitations.

Thank you, we have mentioned this issue among the limitations of the study.

Line 297: Although the acronym QoL is clear, it should be explained. Furthermore, since (like QI), it is used only once, it would be better to replace it with the full term, also in view of the numerous acronyms the authors use in the article.

Thanks for the comment, we corrected it.

Lines 334-336: general statements without adequate supporting bibliographic references.

Thank you, we have followed your suggestion and inserted references to support the sentences.

Line 347: delete the dot at the beginning of the line. It is not clear why the authors refer to "relational needs" only at the end of the article. What is their relationship with the aspects assessed through the totems (punctuality, comfort, PCWP, and professionalism)?  

Thank you for your comment. We have rephrased the whole sentence to focus on the importance of feedback from the cancer patient.

Lines 358-359Another limitation, contrary to the one indicated by the authors, could be that the same person votes twice.

Thank you for pointing this out. We have added this issue within the limits of the study.

Round 2

Reviewer 1 Report

Thanks for your revision

Author Response

Reviewer 1

Comments and Suggestions for Authors

  • Thanks for your revision
  • We thank the reviewer for his/her comment, we provided English language revision as requested.

Reviewer 3 Report

Dear author,

in the following, some minor typing errors need to be edited.

Sentences "The firstly proposed four questions were: “Did you feel welcomed as a person today?” for PCWP; “Did your appointment take place at the scheduled time?” for punctuality; “Have they been professional with you today?” for professionalism and “Are the environments comfortable?” for comfort” are repeated twice (see lines 123-126, and lines 129-133).

In line 138, the authors refer to 1 comfort totem, but in line 148 they refer to 2 comfort totems. Are there 8 (as claimed in line 134) or 9 totems in total?

Table 1 could benefit from some clarifying sentences in its caption.

Line 247: I think that the authors mean “a main” and not amain.

Author Response

Comments and Suggestions for Authors

Dear author,

in the following, some minor typing errors need to be edited.

  • Sentences "The firstly proposed four questions were: “Did you feel welcomed as a person today?” for PCWP; “Did your appointment take place at the scheduled time?” for punctuality; “Have they been professional with you today?” for professionalism and “Are the environments comfortable?” for comfort” are repeated twice(see lines 123-126, and lines 129-133).
  • We thank the reviewer for his/her comment, we checked the manuscript and corrected it.

  • In line 138, the authors refer to 1 comfort totem, but in line 148 they refer to 2 comfort totems. Are there 8 (as claimed in line 134) or 9 totems in total?
  • We thank the reviewer for his/her comment, we checked the manuscript and corrected it.

  • Table 1 could benefit from some clarifying sentences in its caption.
  • We thank the reviewer for his/her comment, we modified Table 1.

  • Line 247: I think that the authors mean “a main” and not amain.
  • We thank the reviewer for his/her comment, we checked the manuscript and corrected it.